# Estimating climate change and mental health impacts in Canada: A cross-sectional survey protocol

Sherilee L. Harper[1]*, Ashlee Cunsolo[2], Breanne Aylward[1], Susan Clayton[3], Kelton Minor[4], Madison Cooper[1], Rachael Vriezen[1]

1 School of Public Health, University of Alberta, Edmonton, Canada, 2 School of Arctic & Subarctic Studies, Labrador Campus of Memorial University, Happy Valley-Goose Bay, Canada, 3 College of Wooster, Wooster, Ohio, United States of America, 4 Data Science Institute, Columbia University, New York, New York, United States of America

* sherilee@ualberta.ca, sherilee.harper@ualberta.ca

**Data Availability Statement:** N/A - No results are reported. This is a protocol paper, so there are no data associated with this paper.

## Abstract

Climate change has severe and sweeping impacts on mental health. Although research is burgeoning on mental health impacts following climate and weather extremes, less is known about how common these impacts are outside of extreme events. Existing research exploring the prevalence of psychosocial responses to climate change primarily examines university students and uses non-random sampling methods. Herein, our protocol outlines an approach to data collection, processing, and analysis to estimate the population prevalence, magnitude, and distribution of mental health responses to climate change in Canada. A cross-sectional survey of youth and adults aged 13 years and older in Canada will be administered over the course of one year. The questionnaire will take approximately 10 minutes to complete orally and will be administered in English, French, and Inuktitut. The survey will consist of six sections: (1) self-reported past experiences of climate change; (2) self-reported climate-related emotions; (3) self-reported past and current impacts, anticipatory impacts, and vicarious experiences; (4) self-reported subclinical outcomes; (5) self-reported behavioural responses; and (6) demographics. A multi-stage, multi-stratified random probability sampling method will be used to obtain a sample representative of the Canadian population. We will use two different modes of recruitment: an addressed letter sent by postal mail or a telephone call (landlines and cellular). Population-weighted descriptive statistics, principal component analysis, and weighted multivariable regression will be used to analyse the data. The results of this survey will provide the first national prevalence estimates of subclinical mental health responses to climate change outcomes of people living in Canada.

## Introduction

Given the mounting evidence, it is undeniable that climate change affects our mental health and wellness [1]. Directly experiencing slow-onset climate changes (e.g. sea level rise, ocean acidification, permafrost thaw) and/or weather and climate extremes (e.g. wildfires, flooding,

**Funding:** Funding support was provided by the Canadian Institutes for Health Research (to Harper and Cunsolo), ArcticNet (to Cunsolo), Social Science and Humanities Research Council Doctoral Fellowship (to Aylward), Izaak Walton Killam Memorial Scholarship (to Aylward), and Alberta Innovates Graduate Student Scholarship (to Aylward). The funders had no role in study design, data collection and analysis, decision to publish, or preparation of the manuscript.

**Competing interests:** The authors have declared that no competing interests exist.

drought, extreme heat waves, storms) has been associated with both diagnosable mental disorders and subclinical outcomes that disrupt or impair normal functioning [1]. Indirectly or vicariously experiencing climate change (e.g. witnessing climate change impacts on the news, seeing climate change affect friends and family, learning about climate change impacts and (in)actions) or anticipating future climate change risks can also lead to mental health challenges [1]. Given these widespread and complex etiological pathways, climate change is increasingly recognized as a grand and urgent challenge for mental health [1–5].

Although we know that climate change has complex, serious, and diverse impacts on mental health, the frequency, magnitude, and distribution of those impacts is less clear. To date, climate change and mental health research has been predominantly quantitative in nature [6–11], and most often examines mental health after a major climate hazard or disaster [6, 9, 12]. For example, many studies have examined hospital admissions for mental health conditions or suicide following extreme heat waves [13, 14], droughts [15], floods [10, 16, 17], and wildfires [18]. Due to the acute nature of these climate events, the association with excess mental health morbidity and mortality is clear and often striking.

However, studies that examine mental health impacts outside of extreme events are less common [6, 9]. Further, previous studies that explored climate change impacts on psychosocial wellbeing have primarily relied on surveys using non-probability sampling with university students and have often used crowdsourcing (e.g. Amazon MTurk) to source participants [19]. Both of these methods can create study-entry selection bias due to non-random sample selection [20], particularly coverage bias and self-selection bias [21], which can result in a study sample that does not represent the target population and compromises the external validity of results. New climate and mental health survey literature seeks to address these potential biases by deploying systematically inclusive national surveys that aim to provide every member of the population with an equal chance of being included [22].

There is also a small but growing body of qualitative research exploring climate change impacts on psychosocial wellbeing. Qualitative research has provided insights into the diverse pathways and mechanisms through which climate change affects mental health [23, 24]. For example, in-depth, predominantly qualitative research has illustrated how climate change has led to ecological grief for Inuit in Arctic Canada and for farmers in the Australian Wheatbelt [25]. A growing number of qualitative studies have also documented experiences of solastalgia, or the pain caused by negative environmental changes to one's home [26], especially among people in the United States and Australia [27]. Although these studies provide detailed insights into the ways in which climate change affects mental health, they do not provide comprehensive data on how common these climate-mental health experiences are in the general population.

Canada is a global leader in advancing our understanding of climate change impacts on mental health. Indeed, a relatively large proportion of published research on this topic has come from Canada [6, 7, 28], particularly qualitative studies [6]. Evidence on the mental health implications of climate change is beginning to inform policy, programs, and decision-making in Canada. For example, mental health has been included—for the first time—in Canada's national climate change assessments [e.g. 3, 29, 30], in Canada's first National Adaptation Strategy [31], and in the Chief Public Health Officer's 2022 annual report [32].

The available evidence clearly suggests that climate change has substantial implications for mental health, and there is increasing recognition of the importance of these impacts in policy decision-making in Canada. However, when sharing this research evidence with decision-makers, policy-makers, and the general public, our research team is often asked about the national prevalence and distribution of climate change impacts on mental health, and about the diverse pathways through which climate change negatively affects mental health across the

country. Currently, we do not have a comprehensive estimate of the distribution and prevalence of the mental health impacts of climate change in Canada, and little evidence is available globally. Therefore, our team has developed a protocol for a nationally representative cross-sectional survey to estimate the prevalence, magnitude, and distribution of mental health challenges connected to climate change in Canada. Herein, we describe the protocol for data collection, processing, and analysis. Although this protocol has been developed for use in Canada, the study design and methods aim to overcome common challenges and limitations in this field of research, and our call for epidemiological rigour in climate-mental health survey design is applicable globally.

## Methods

### Primary objectives

The overall goal of the study is to estimate the prevalence of a variety of emotional responses and subclinical mental health outcomes that disrupt or impair normal functioning as they relate to climate change (Table 1). Specifically, the primary objective is to estimate the prevalence of climate-related emotions and subclinical outcomes by province/territory, by month, and by age group (i.e. 13–34 years; 35–54 years; 55–74 years; 75+ years old). Secondary objectives include examining the association of subclinical outcomes with various socio-demographic variables (e.g. age, gender, income, and other identities), direct or vicarious experiences of past or current climate change impacts, anticipation of future climate change impacts, and behavioural responses to climate-related stressors.

### Design

This population-based cross-sectional survey (i.e. prevalence study) of youth and adults aged 13 years and older in Canada will be administered over the course of one year (12 months). The protocol was developed by an interdisciplinary team of researchers in Canada and the USA with expertise spanning mental health, climate change, psychology, epidemiology, social sciences, health geography, economics, and public health. The survey was administered by R. A. Malatest & Associates Ltd. (Malatest), one of Canada's largest independently owned and operated research and evaluation firms. Survey recruitment commenced on April 1, 2022, and data collection is expected to continue through to March 31, 2023.

### Climate change and mental health scales and theory

Climate change anxiety is a psychological phenomenon that has gained prominence in recent years. Clayton and Karaszia (2020) developed and validated a scale that quantifies the extent to which individuals experience anxiety related to climate change [33]. Their analysis found that

**Table 1. A summary of research questions related to our primary survey objective.**

| Primary Research Question | Example Question |
| --- | --- |
| RQ1 | What is the prevalence of climate-related emotions and subclinical outcomes in Canada? |
| RQ2 | What is the prevalence of climate-related emotions and subclinical outcomes in each province/territory in Canada? |
| RQ3 | What is the prevalence of climate-related emotions and subclinical outcomes in Canada each month? |
| RQ4 | What is the prevalence of climate-related emotions and subclinical outcomes in Canada by age group? |

climate change anxiety was made up of two latent constructs: cognitive emotional impairment, which encompassed rumination, sleep-related concerns, difficulties concentrating, and crying, and functional impairment, which encompassed impairments in day-to-day functioning, including tasks at work or school. Some subsequent studies validated the original two-factor structure of the climate change and anxiety scale in other countries and languages [e.g., 34, 35], whereas others could not replicate the two-factor structure [e.g., 36, 37]. Wullenkord et al. (2021) could not reproduce the two-factor structure in the German population, leading them to question the construct validity of the climate change anxiety scale [37]. They recommended further scale development to encompass varying levels of intensity of climate change anxiety, as well as other related emotional experiences. Specifically, they argued for incorporating an emotional factor to measure feelings associated with anxiety (e.g., worry and fear) and for adding items that capture experiences of uncertainty that are traditionally associated with anxiety. Therefore, in our study, we incorporated Clayton and Karasizia's (2020) climate change anxiety scale [33] as well as Searle and Gow's (2010) [38] scale of climate-related emotions, and we also selected items from Reser et al. (2012) that measure feelings of unpredictability related to climate change [43] (Table 2).

## Survey instrument

The survey consists of six sections: (1) self-reported past experiences of climate change; (2) self-reported climate-related emotions; (3) self-reported past and current impacts, anticipatory impacts, and vicarious experiences of climate change; (4) self-reported subclinical mental health outcomes; (5) self-reported behavioural responses; and (6) demographics (Table 2). The questionnaire was written in English and translated into French and Inuktitut. Translations were back-translated into English to ensure accuracy. The questionnaire was designed to take approximately 10 minutes to complete over the telephone. The English version of the full questionnaire is available in S1 Table.

Over a one-week period, we used a convenience sample (n = 35) to pretest the online questionnaire and a random sample (n = 20) to pretest the telephone questionnaire for length and content. We refined the survey based on estimated completion times as well as respondent feedback. The data collected during pretesting will not be included in the final dataset.

## Eligible participants

The questionnaire will be administered in all provinces and territories in Canada. Youth and adults who are 13 years old or older and have a Canadian postal address or telephone number will be eligible to participate. The survey will be offered in English, French, or Inuktitut. Attempts will be made to accommodate participants that speak languages other than English, French, or Inuktitut on an as-requested basis.

## Sampling

**Sample size.** There are no published prevalence estimates of climate change-related mental health outcomes that are representative of Canada. Therefore, we calculated our target sample sizes using an *a priori* estimate of 25% prevalence of subclinical outcomes related to climate change based on estimates from the United States of America [33] to calculate a target sample size of 289 people per province/territory. We used an allowable error of 5% and a confidence level of 95% to calculate a target sample size of 289 people per province/territory. Reflecting our objective to estimate prevalence by province/territory, our total target sample size is therefore 3,757 people [i.e. (289 people per province or territory) x (10 provinces + 3 territories) = 3,757 people]. Then, reflecting our objective to estimate the national prevalence by

**Table 2. A summary of the questionnaire sections, with a description of the items in the section, an overview of the purpose of the section, and information on how the items were developed.**

| Questionnaire Section | Goal of Questions | Measures | Validity |
|---|---|---|---|
| **1) Past experiences of climate change** | To capture information about individual experiences of climate change impacts, which can be a precursor to subclinical mental health outcomes. | 2 x 5-point Likert scale questions capturing direct and indirect exposures to climate change. | We selected two of the three questions assessing personal experiences of climate change validated by Clayton and Karazsia [33]. |
| **2) Climate-related emotions** | To capture a range of subclinical outcomes and emotions that can be experienced in response to climate change. | 13 x 5-point Likert scale questions capturing emotional responses to climate change (i.e. sad, angry, grief, helpless, hopeless, worried, anxious, depressed, tense, concerned, stressed, scared, and powerless). | Our questions cover the 12 emotions assessed by Searle and Gow [38], and we added an additional question about grief. |
| **3) Past and current impacts, anticipatory impacts, and vicarious experiences** | To measure the drivers of respondents' emotional reactions to climate change. | 3 x 5-point Likert scale questions capturing the extent to which respondents' reactions to climate change were related to past and current climate change impacts, anticipated future climate change impacts, or news and social media coverage. | We developed new questions based on the identified pathways of ecological grief defined by Cunsolo and Ellis [25]. |
| **4) Subclinical outcomes** | To measure cognitive and emotional impairment from climate change exposures. | 8 x 5-point Likert scale questions capturing climate change impacts on concentration and rumination. | We adapted the cognitive and emotional impairment questions in Clayton and Karazsia's [33] climate change anxiety scale. This scale has been used and/or validated in subsequent studies by Cruz and High [39], Innocenti et al. [36], Larionow et al. [34], Mouguima-Daouda et al. [35], Simon et al. [40], and Joo Jang et al. [41]. |
|  | To measure functional impairment from climate change exposures. | 1 x 5-point Likert scale question capturing climate change impacts on daily life. | We adapted the five functional impairment questions from Clayton and Karazsia's [33] climate anxiety scale to create one proxy question. Hickman et al. [42] also used this question as a proxy for functional impairment. |
|  | To measure other climate-related worry, thinking, and distress. | 5 x 5-point Likert scale questions capturing feelings of uncertainty in response to climate change. | We selected and adapted questions from Reser et al.'s [43] 12-item, 6-point scale to address Wullenkord et al.'s [37] critiques of Clayton and Karazsia's [33] climate anxiety scale. |
| **5) Behavioural responses** | To assess whether climate change anxiety is associated with individual or collective climate actions. | 3 x 5-point Likert scale questions capturing individual actions in response to climate change, and 3 x 5-point Likert scale questions capturing collective actions in response to climate change. | We selected and adapted questions from Stanley et al.'s [44] 16-item, 100-point scale. One of these questions was similar in concept to Clayton and Karazsia's [33] 5-item measure of sustainable behaviour. |
| **6) Demographics** | To collect independent variables for use in subsequent analyses. | 5 close-ended questions capturing age, gender identities, ethnic backgrounds, income, and postal code. | Questions were based on the Alberta Human Rights Commission's [45] description of gender identity, the Government of Canada's Employment Equity Act's definition of visible minority groups [46], and the Public Health Agency of Canada's Foodbook Questionnaire (e.g. for income) [47]. |

month, we distributed the target sample of 3,757 people over 12 months; that is, our monthly target sample size is 313 people per month [i.e. (3,757 people) / (12 months) = 313 people per month] or 24 people per province/territory per month [i.e. (313 people per month) / (13 provinces or territories) = 24 people per province or territory per month].

**Sampling procedure.** A multi-stage, multi-stratified random probability sampling method will be used to obtain a sample representative of Canada (Fig 1). Participants will be randomly selected via a two-stage approach. First, households will be randomly selected and recruited either by an addressed letter or a telephone call (landlines and cellular). Then, within each randomly selected household, the person who is 13 years old or older and has the next birthday will be invited to participate in the survey.

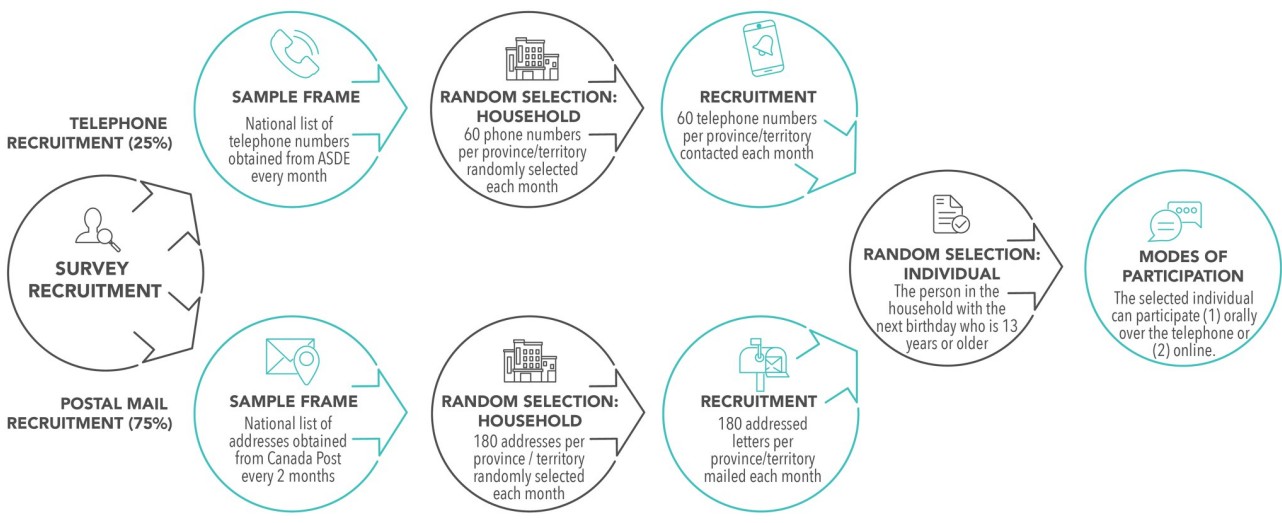

**Fig 1. An overview of the multi-stage random sampling method.**

To develop a sampling frame for mail recruitment, we will draw from a national list of addresses maintained by Canada Post. Canada Post will provide a new sampling frame every two months by randomly selecting addresses to meet our sample parameters. From this sampling frame, we will randomly select addresses and we will mail survey invitations as addressed letters to those addresses. Addressed letters will be sent monthly and will contain instructions for completing the questionnaire.

To develop a sampling frame for telephone recruitment, we will obtain a national list of landline and cellular telephone numbers from ASDE Inc., a survey sample company that purchases this information from phone companies. A new sampling frame will be developed each month.

## Recruitment

We will use two different modes of recruitment: an addressed letter sent by postal mail and a telephone call (landlines and cellular). Ideally, participants will only be contacted by one mode; however, some overlap between mail and telephone sampling is theoretically possible as these samples will be drawn from separate but not mutually exclusive databases, although any overlap is statistically improbable. The modes of recruitment will be initially allocated as follows: 75% for mail recruitment and 25% for telephone recruitment.

Participants recruited via addressed letter will be directed to a website containing the online questionnaire, or they can phone (toll-free) to complete the questionnaire orally. Initially, we will mail 180 addressed letters in each province and territory, assuming an estimated 10% response rate. We will monitor the response rates for each province and territory, as well as for urban and rural areas using the first two digits of respondents' postal codes, and we will adjust the sample parameters and the number of randomly selected addresses accordingly. Each selected household will only receive one addressed letter during the study period, with the exception of households in Nunavut. In all other provinces and territories, our address-based sample will include a household address and corresponding telephone number; thus, we will be able to follow up on mailed survey invitations with a reminder telephone call, which will be made two weeks after the addressed letters are mailed, provided that selected participants have not completed the survey. Reminder telephone calls will not be possible for households in

Nunavut, since each household address in the Canada Post database does not include a corresponding telephone number. Instead, we will send one additional reminder letter to random households that have not completed the survey.

Initially, the sampling frame for telephone recruitment will include 60 telephone numbers per province/territory, assuming a 10% response rate. Similar to our approach to our mail-based sampling frame, we will monitor the response rates for each province and territory, as well as for urban and rural areas, and we will make adjustments to the sample parameters and the number of randomly selected telephone numbers accordingly. Within the first 10 days of each month, we will dial each of the selected phone numbers at least once. Phone numbers will be dialled on the following days and times in the time zone of the area being surveyed: Monday to Friday between 13:00 to 20:00; Saturday between 10:00 to 18:00; and Sunday between 12:00 to 20:00. No initial calls will be made on statutory holidays. Each call attempt will allow for a minimum of five rings or until it is sent to voicemail. Initially, we will attempt each telephone number up to five times over the course of one month. Each call will be made on different days of the week and during different times of the day. A voicemail message will be left on the second unsuccessful attempt and every unsuccessful attempt thereafter. Eligible participants can complete the questionnaire over the telephone with a trained interviewer; schedule an appointment to be called back later; or receive a link by email or text message to complete the questionnaire online. All callback appointments will be scheduled for the day and time requested by the participant.

Regardless of the recruitment method (mail or telephone), participants will complete the same questionnaire either through Voxco via Computer Assisted Telephone Interviewing (CATI) or using an online survey instrument. However, the introductory recruitment script will differ depending on the mode of recruitment: the telephone version requires more introductory information as well as the questions that are necessary to randomly select the member of the household to interview (i.e. the person 13 years or older in the household who has the next birthday). For both modes of recruitment, the day that the questionnaire is completed will be logged as the reference day (i.e. the day when the last question was completed and the survey was submitted).

To achieve a representative sample, we will aim for the number of completed questionnaires to be proportional to the distribution of age groups (13–34 years; 35–54 years; 55–74 years; 75 + years) in each province/territory based on Statistics Canada data (2021) [48] (Table 3). Additionally, within each province/territory, we will aim for the number of questionnaires completed to be proportional to that province/territory's urban/rural population sizes based on the first two digits of respondents' postal codes based on Statistics Canada data (2016) [49].

We will use the Market Research and Intelligence Agency (MRIA) empirical method to calculate the questionnaire response rate [50]. Thus, we will divide the number of responding units by the sum of in-scope and unresolved units. The MRIA empirical method is widely used in Canada, including in previous surveys conducted by the Government of Canada [47, 50].

## Options to overcome recruitment challenges

If any age or geographical targets are not met during the course of the survey administration, we will employ several methods to increase recruitment success. For example, the proportion of cellular phone, landline telephone, and mailout recruitments in the sample can be adjusted. For those participants contacted by cellular telephone, if there is no response after the third attempt, one text message may be sent to the cellular phone containing a link to the online version of the questionnaire. Small incentives (i.e. gift cards) may be used to increase enrolment for subpopulations that might have lower response rates (e.g. those living in the Territories

**Table 3. A summary of the target number of responses by province/territory and age group for the entire survey period (1 year).**

| Provinces/Territories | Target Number of Respondents by Age Groups | | | | |
|---|---|---|---|---|---|
| | 13–34 years | 35–54 years | 55–74 years | 75+ years | Total* |
| Alberta | 100 | 97 | 72 | 20 | 289 |
| British Columbia | 91 | 87 | 84 | 27 | 289 |
| Manitoba | 103 | 86 | 76 | 24 | 289 |
| New Brunswick | 80 | 83 | 96 | 30 | 289 |
| Newfoundland and Labrador | 77 | 84 | 99 | 29 | 289 |
| Northwest Territories | 112 | 98 | 70 | 9 | 289 |
| Nova Scotia | 86 | 81 | 93 | 29 | 289 |
| Nunavut | 144 | 97 | 43 | 5 | 289 |
| Ontario | 97 | 86 | 80 | 26 | 289 |
| Prince Edward Island | 94 | 81 | 87 | 27 | 289 |
| Quebec | 86 | 87 | 87 | 29 | 289 |
| Saskatchewan | 99 | 88 | 77 | 25 | 289 |
| Yukon | 95 | 99 | 81 | 14 | 289 |
| Total* | 1,214 | 1,135 | 1,068 | 340 | 3,757 |

*Target number of respondents were calculated to be proportional to the age demographics of that province or territory using data from Statistics Canada (2021) [48] and may differ from the cumulative sums of each column due to rounding.

and younger age groups). If the subpopulation is small, targeted social media advertisements may be used to increase awareness of the survey (but note that these advertisements will not be used to facilitate recruitment). Quotas for certain demographic characteristics can be set each month; that is, once a quota has been met for a demographic group, the questionnaire will not be available for participants belonging to that group for the remainder of the month. Finally, to help reach our demographic quotas, we could limit or remove the ability for certain demographics to be recruited over the telephone (i.e. those in a certain age group are not eligible to be recruited via telephone).

## Ethics

All participants will be provided with an information letter that outlined the study details. We will obtain informed consent from all participants: telephone participants provided informed consent orally (documented by the surveyor) and online participants provided informed consent electronically. We were granted a waiver of parental/guardian consent for participants between the ages of 13–18, given that our study met all criteria under Article 3.7 of the TCPS2. Our study is an anonymous survey that collects minimal identifying information and involves minimal risk to participants. The study protocol was approved by the ethics research boards at the University of Alberta and Memorial University of Newfoundland and Labrador. A research licence was obtained from the Nunavut Research Institute.

## Data processing and analyses

The questionnaire response data will be downloaded in English (all responses to open-ended questions in languages other than English will be translated into English by Malatest and then provided to the research team) into Excel (Microsoft Corporation, Redmond, WA), and then uploaded into Stata (StataCorp LLC, College Station, TX) and R (The R Foundation, Indianapolis, IN) for analysis. Data will be population-weighted (e.g. by age, province/territory, gender,

rural/remote, and/or income based on Statistics Canada data) to account for oversampling and for the biases inherent in the selection strategy. Descriptive statistics will be calculated and will reflect the survey objectives. The prevalence of climate-related subclinical mental health outcomes and emotions and the measures of central tendency for the independent variables will be calculated. Outcome measures (prevalence) and differences between demographic groups will be calculated with 95% confidence intervals. Factor analyses will be conducted for the emotion and subclinical outcome questions [33, 38]. Population-weighted multivariable regression analyses will be used to explore associations between the outcome variables and the independent variables.

## Discussion

This study will provide the first national prevalence estimates of climate change impacts on psychosocial wellbeing in Canada. By examining the prevalence, magnitude, and distribution of climate-sensitive mental health impacts in Canada, this survey will provide important national evidence to support decision-making, adaptation strategies, and public health investments. In addition, although this survey protocol has been developed for use in Canada, the overall design and methods can serve as a foundation for further understanding climate-related mental health outcomes that can inform and support other national and international surveys on this important topic. Finally, the results of this survey will add population-level insights to the global climate change and mental health knowledge base.

## Supporting information

**S1 Table. English version of the full questionnaire.**
(DOCX)

## Acknowledgments

We extend our thanks to Nia King and Hannah Bayne for translation support. Special thanks to Hannah Bayne and Kelsey Robertson for their support in developing elements of the questionnaire and pre-testing phases. We sincerely thank Doug Elliot from Malatest for his ongoing support and troubleshooting ability.

## Author Contributions

**Conceptualization:** Sherilee L. Harper, Ashlee Cunsolo, Breanne Aylward, Susan Clayton, Kelton Minor, Madison Cooper, Rachael Vriezen.

**Data curation:** Breanne Aylward.

**Funding acquisition:** Sherilee L. Harper, Ashlee Cunsolo.

**Methodology:** Sherilee L. Harper, Ashlee Cunsolo, Breanne Aylward, Susan Clayton, Kelton Minor, Madison Cooper, Rachael Vriezen.

**Resources:** Sherilee L. Harper.

**Visualization:** Sherilee L. Harper, Ashlee Cunsolo.

**Writing – original draft:** Sherilee L. Harper.

**Writing – review & editing:** Sherilee L. Harper, Ashlee Cunsolo, Breanne Aylward, Susan Clayton, Kelton Minor, Madison Cooper, Rachael Vriezen.

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
