## [Decision Letter · Decision Letter 0]

20 Jul 2023

PONE-D-23-09222Estimating climate change and mental health impacts in Canada: A cross-sectional survey protocolPLOS ONE

Dear Dr. Harper,

Thank you for submitting your manuscript to PLOS ONE. After careful consideration, we feel that it has merit but does not fully meet PLOS ONE’s publication criteria as it currently stands. Therefore, we invite you to submit a revised version of the manuscript that addresses the points raised during the review process.

We look forward to receiving your revised manuscript.

Kind regards,

Dr. Ulysses Paulino Albuquerque

Academic Editor

PLOS ONE

Journal Requirements:

2. In the ethics statement in the Methods, you have specified that verbal consent was obtained. Please provide additional details regarding how this consent was documented and witnessed, and state whether this was approved by the IRB

“Funding support was provided by CIHR (to Harper and Cunsolo), ArcticNet (to Cunsolo), SSHRC Doctoral Fellowship (to Aylward), Izaak Walton Killam Memorial Scholarship (to Aylward), and Alberta Innovates Graduate Student Scholarship (to Aylward).”

4. Please expand the acronym “CIHR and SSHRC” (as indicated in your financial disclosure) so that it states the name of your funders in full.

“Funding support was provided by CIHR (to Harper and Cunsolo), ArcticNet (to Cunsolo), SSHRC Doctoral Fellowship (to Aylward), Izaak Walton Killam Memorial Scholarship (to Aylward), and Alberta Innovates Graduate Student Scholarship (to Aylward).”

“Funding support was provided by CIHR (to Harper and Cunsolo), ArcticNet (to Cunsolo), SSHRC Doctoral Fellowship (to Aylward), Izaak Walton Killam Memorial Scholarship (to Aylward), and Alberta Innovates Graduate Student Scholarship (to Aylward).”

Additional Editor Comments:

After carefully reviewing your article, I would like to provide some guidance to enhance the quality of your work. Specifically, the introduction of your study needs to be reformulated to include clear hypotheses and predictions for the proposed research protocol. Furthermore, it is essential to substantiate these predictions based on the available evidence in the relevant literature.

An adequate introduction should provide a literature review, establishing a solid theoretical foundation for the study at hand. It is crucial for you to identify existing knowledge gaps and present plausible hypotheses to address these gaps. These hypotheses should be formulated in a clear and straightforward manner, demonstrating a strong understanding of the research problem.

Additionally, I suggest that you include a power analysis to estimate the required sample size for your study. Power analysis is an important statistical tool that allows determining the appropriate sample size to achieve significant and reliable results. Including this analysis in your article will strengthen the validity of your findings.

When conducting the power analysis, it is crucial to consider various factors such as the expected effect size, desired significance level, desired statistical power, and data variability. This analysis will enable you to determine the minimum number of participants needed for the study, taking into account the statistical sensitivity of your protocol.

I emphasize the importance of substantiating your predictions and hypotheses based on the available evidence. This will ensure that your study is grounded on a solid foundation, contributing to the reliability and validity of the obtained results.

Reviewers' comments:

Reviewer's Responses to Questions

**Comments to the Author**

1. Does the manuscript provide a valid rationale for the proposed study, with clearly identified and justified research questions?

Reviewer #1: Yes

2. Is the protocol technically sound and planned in a manner that will lead to a meaningful outcome and allow testing the stated hypotheses?

Reviewer #1: Yes

3. Is the methodology feasible and described in sufficient detail to allow the work to be replicable?

Reviewer #1: Yes

4. Have the authors described where all data underlying the findings will be made available when the study is complete?

Reviewer #1: Yes

5. Is the manuscript presented in an intelligible fashion and written in standard English?

Reviewer #1: Yes

6. Review Comments to the Author

You may also provide optional suggestions and comments to authors that they might find helpful in planning their study.

Reviewer #1: I want to congratulate the authors on their submitted protocol and for the detailed description of the procedure to be used in carrying out the proposed study.

I believe that the procedure adequately describes all the steps necessary to perform and analyze the data of the study.

7. PLOS authors have the option to publish the peer review history of their article (what does this mean?). If published, this will include your full peer review and any attached files.

Reviewer #1: **Yes: **Mauro Dias Silva Júnior

---

## [Author Response · Author response to Decision Letter 0]

9 Aug 2023

Responses to Editor Comments

Editor Comment: 1. Please ensure that your manuscript meets PLOS ONE's style requirements, including those for file naming. The PLOS ONE style templates can be found at

Authors’ response: We have checked the style requirements, and we believe that we have met all of the criteria. If we have missed something, let us know and we will correct it right away.

Editor Comment: 2. In the ethics statement in the Methods, you have specified that verbal consent was obtained. Please provide additional details regarding how this consent was documented and witnessed, and state whether this was approved by the IRB

Author’s response: Thank you for this comment. We have clarified how this consent was documented (new text underlined):

“We will obtain informed consent from all participants: telephone participants provided informed consent orally (documented by the surveyor) and online participants provided informed consent electronically.”

Ethics approval from several ethics boards (IRBs) for our entire procedure. This is stated in the following sentence in the methods section:

“The study protocol was approved by the ethics research boards at the University of Alberta and Memorial University of Newfoundland and Labrador. A research licence was obtained from the Nunavut Research Institute.”

Editor Comment: 3. Thank you for stating the following financial disclosure: “Funding support was provided by CIHR (to Harper and Cunsolo), ArcticNet (to Cunsolo), SSHRC Doctoral Fellowship (to Aylward), Izaak Walton Killam Memorial Scholarship (to Aylward), and Alberta Innovates Graduate Student Scholarship (to Aylward).” Please state what role the funders took in the study. If the funders had no role, please state: "The funders had no role in study design, data collection and analysis, decision to publish, or preparation of the manuscript." If this statement is not correct you must amend it as needed. Please include this amended Role of Funder statement in your cover letter; we will change the online submission form on your behalf.

Authors’ response: Thank you for this comment. We have added the following text to our cover letter, to the online submission form, and in our manuscript (new text is underlined): 

Editor Comment: 4. Please expand the acronym “CIHR and SSHRC” (as indicated in your financial disclosure) so that it states the name of your funders in full.

Authors’ response: We are happy to confirm that we have included this text to our cover letter, and corrected it int the online submission form and in our manuscript.

Editor Comment: 5. We note that the grant information you provided in the ‘Funding Information’ and ‘Financial Disclosure’ sections do not match. When you resubmit, please ensure that you provide the correct grant numbers for the awards you received for your study in the ‘Funding Information’ section.

Authors’ response: Unfortunately, I did not see a section in the submission portal entitled “Funding Information”, so we were unable to see what error we made and make any corrections needed. We are happy to provide any clarifications or make corrections. Please note that our funders listed do not have “grant numbers”, so we would list “n/a” under any line in the online portal asking for “grant numbers”.

Editor Comment: 6. Thank you for stating the following in the Acknowledgments Section of your manuscript: “Funding support was provided by CIHR (to Harper and Cunsolo), ArcticNet (to Cunsolo), SSHRC Doctoral Fellowship (to Aylward), Izaak Walton Killam Memorial Scholarship (to Aylward), and Alberta Innovates Graduate Student Scholarship (to Aylward).” We note that you have provided additional information within the Acknowledgements Section that is not currently declared in your Funding Statement. Please note that funding information should not appear in the Acknowledgments section or other areas of your manuscript. We will only publish funding information present in the Funding Statement section of the online submission form. Please remove any funding-related text from the manuscript and let us know how you would like to update your Funding Statement. Currently, your Funding Statement reads as follows: “Funding support was provided by CIHR (to Harper and Cunsolo), ArcticNet (to Cunsolo), SSHRC Doctoral Fellowship (to Aylward), Izaak Walton Killam Memorial Scholarship (to Aylward), and Alberta Innovates Graduate Student Scholarship (to Aylward).” Please include your amended statements within your cover letter; we will change the online submission form on your behalf.

Authors’ response. Thank you for this comment. We have removed this information from the manuscript, and included it in our cover letter, as requested. The funding statement should read:

Funding support was provided by the Canadian Institutes for Health Research (to Harper and Cunsolo), ArcticNet (to Cunsolo), Social Science and Humanities Research Council Doctoral Fellowship (to Aylward), Izaak Walton Killam Memorial Scholarship (to Aylward), and Alberta Innovates Graduate Student Scholarship (to Aylward). The funders had no role in study design, data collection and analysis, decision to publish, or preparation of the manuscript.

Editor Comment: After carefully reviewing your article, I would like to provide some guidance to enhance the quality of your work. Specifically, the introduction of your study needs to be reformulated to include clear hypotheses and predictions for the proposed research protocol. Furthermore, it is essential to substantiate these predictions based on the available evidence in the relevant literature.

Authors’ response: Thank you for these thoughtful comments. We have addressed each of these comments, in detail, below. 

Editor Comment: An adequate introduction should provide a literature review, establishing a solid theoretical foundation for the study at hand. It is crucial for you to identify existing knowledge gaps and present plausible hypotheses to address these gaps. These hypotheses should be formulated in a clear and straightforward manner, demonstrating a strong understanding of the research problem.

Authors’ response: Thank you for this comment. We have added a new section in the methods section, in order to expand on the background we provided in the introduction. In this new section, we outline the theoretical foundation of the scales that we used in our survey (new text is underlined):

“Climate change and mental health scales and theory

Climate change anxiety is a psychological phenomenon that has gained prominence in recent years. Clayton and Karaszia (2020) developed and validated a scale that quantifies the extent to which individuals experience anxiety related to climate change [33]. Their analysis found that climate change anxiety was made up of two latent constructs: cognitive emotional impairment, which encompassed rumination, sleep-related concerns, difficulties concentrating, and crying, and functional impairment, which encompassed impairments in day-to-day functioning, including tasks at work or school. Some subsequent studies validated the original two-factor structure of the climate change and anxiety scale in other countries and languages [e.g., 34,35], whereas others could not replicate the two-factor structure [e.g., 36,37]. Wullenkord et al. (2021) could not reproduce the two-factor structure in the German population, leading them to question the construct validity of the climate change anxiety scale [37]. They recommended further scale development to encompass varying levels of intensity of climate change anxiety, as well as other related emotional experiences. Specifically, they argued for incorporating an emotional factor to measure feelings associated with anxiety (e.g., worry and fear) and for adding items that capture experiences of uncertainty that are traditionally associated with anxiety. Therefore, in our study, we incorporated Clayton and Karasizia’s (2020) climate change anxiety scale [33] as well as Searle and Gow’s (2010) [38] scale of climate-related emotions, and we also selected items from Reser et al. (2012) that measure feelings of unpredictability related to climate change [42] (Table 1).”

We did not include hypotheses in our manuscript, as the objective of our study is to estimate prevalence. In a prevalence study there is no hypothesis to test (e.g. Swinscow & Campbell (1997). Study design and choosing a statistical test. BMJ). But, to improve clarity of this in our manuscript, we have moved the section outlining our “Primary Objectives” up in the methods section: it is now the first paragraph in our methods section. We have also added a new table, which outlines pre-specified “Research Questions” related to our primary objective. We hope that this new placement of text helps make this clearer to the reader.

Editor Comment: Additionally, I suggest that you include a power analysis to estimate the required sample size for your study. Power analysis is an important statistical tool that allows determining the appropriate sample size to achieve significant and reliable results. Including this analysis in your article will strengthen the validity of your findings. When conducting the power analysis, it is crucial to consider various factors such as the expected effect size, desired significance level, desired statistical power, and data variability. This analysis will enable you to determine the minimum number of participants needed for the study, taking into account the statistical sensitivity of your protocol.

Authors’ response: Thank you for this suggestion. Since our primary objective is to estimate prevalence, we calculated a sample size. In our sample size calculation, we used an allowable error of 5% and a confidence level of 95% to calculate a target sample size. The details of this calculation were previously in a footnote. However, to improve clarity of this information, we have moved this text out of the footnote and into the main text. We hope that this clarifies how power considerations were approached in our prevalence study design.

Editor Comment: I emphasize the importance of substantiating your predictions and hypotheses based on the available evidence. This will ensure that your study is grounded on a solid foundation, contributing to the reliability and validity of the obtained results.

Authors’ response: Thank you for this comment. We have added a new section in the methods section, in order to expand on the background we provided in the introduction. In this new section, we outline the theoretical foundation of the scales that we used in our survey (new text is underlined):

“Climate change and mental health scales and theory

Climate change anxiety is a psychological phenomenon that has gained prominence in recent years. Clayton and Karaszia (2020) developed and validated a scale that quantifies the extent to which individuals experience anxiety related to climate change [33]. Their analysis found that climate change anxiety was made up of two latent constructs: cognitive emotional impairment, which encompassed rumination, sleep-related concerns, difficulties concentrating, and crying, and functional impairment, which encompassed impairments in day-to-day functioning, including tasks at work or school. Some subsequent studies validated the original two-factor structure of the climate change and anxiety scale in other countries and languages [e.g., 34,35], whereas others could not replicate the two-factor structure [e.g., 36,37]. Wullenkord et al. (2021) could not reproduce the two-factor structure in the German population, leading them to question the construct validity of the climate change anxiety scale [37]. They recommended further scale development to encompass varying levels of intensity of climate change anxiety, as well as other related emotional experiences. Specifically, they argued for incorporating an emotional factor to measure feelings associated with anxiety (e.g., worry and fear) and for adding items that capture experiences of uncertainty that are traditionally associated with anxiety. Therefore, in our study, we incorporated Clayton and Karasizia’s (2020) climate change anxiety scale [33] as well as Searle and Gow’s (2010) [38] scale of climate-related emotions, and we also selected items from Reser et al. (2012) that measure feelings of unpredictability related to climate change [42] (Table 1).”

Reviewers’ comments:

Reviewer #1: I want to congratulate the authors on their submitted protocol and for the detailed description of the procedure to be used in carrying out the proposed study. I believe that the procedure adequately describes all the steps necessary to perform and analyze the data of the study.

Authors’ response: We kindly thank the reviewer for their encouraging comments.

---

## [Editor Report · Decision Letter 1]

29 Aug 2023

Estimating climate change and mental health impacts in Canada: A cross-sectional survey protocol

PONE-D-23-09222R1

Dear Dr. Harper,

We’re pleased to inform you that your manuscript has been judged scientifically suitable for publication and will be formally accepted for publication once it meets all outstanding technical requirements.

Kind regards,

Ulysses Paulino Albuquerque

Academic Editor

PLOS ONE
---

## [Editor Report · Acceptance letter]

2 Oct 2023

PONE-D-23-09222R1 

Estimating climate change and mental health impacts in Canada: A cross-sectional survey protocol 

Dear Dr. Harper:

I'm pleased to inform you that your manuscript has been deemed suitable for publication in PLOS ONE. Congratulations! Your manuscript is now with our production department. 

Kind regards, 

on behalf of

Dr. Ulysses Paulino Albuquerque 

Academic Editor

PLOS ONE